# Sustainable Sheep and Goat Farming in Arid Regions of Jordan

**Faisal Al-Barakeh [1], Ashraf Omar Khashroum [2], Radi A. Tarawneh [3], Fatima A. Al-Lataifeh [4], Azzam N. Al-Yacoub [5], Moammar Dayoub [6] and Khaled Al-Najjar [7,*]**

1 Livestock Research Directorate, National Agriculture Research Center, Amman 11118, Jordan; falbarakeh@yahoo.com
2 Department of Plant Production and Protection Environment, Faculty of Agriculture, Jerash University, Jerash 26150, Jordan; askshb@yahoo.com
3 Department of Economics, Faculty of Agriculture, Jerash University, Jerash 26150, Jordan; radi.amtarawneh@yahoo.com
4 Department of Animal Production and Protection, Faculty of Agriculture, Jerash University, Jerash 26150, Jordan; fatima.allataifah@jpu.edu.jo
5 Department of Biology, Faculty of Applied Science, Umm Al-Qura University, Makkah 24382, Saudi Arabia; azznar@hotmail.com
6 Department of Computing, Faculty of Technology, University of Turku, 20014 Turku, Finland; moammar.dayoub@utu.fi
7 General Commission for Scientific Agricultural Research, GCSAR, Damascus 12573, Syria
* Correspondence: khnajj2011@yahoo.com

**Simple Summary:** This study explores the challenges faced by small farmers in Jordan's dry areas when raising sheep and goats, focusing on limited fodder and water, to find solutions to help farmers maintain healthy and sustainable herds. Despite the obstacles, sheep and goat farming could thrive with factors like breeder experience and economic stability. Crop cultivation and proper winter care are vital to reducing costs and preventing animal mortality. Effective management and support offer opportunities for rearing enhancement. The conclusions provide crucial guidance for supporting small-scale farmers, presenting actionable strategies to address resource limitations. Implementing these recommendations could empower farmers, improve practices, and contribute to economic stability. This study is valuable for policymakers and agricultural organizations promoting sustainable rural development and livelihood enhancement in drylands.

**Abstract:** This study examines the sustainability of livestock farming in Jordan's arid regions, focusing on smallholder Awassi sheep and Baladi goat farming. It assesses breeders, economic factors, herd productivity, and sustainability, evaluating the vitality of these breeds by examining reproductive success through birth and survival rates. In 2021–2022, a total of 53 traditional Awassi sheep and Baladi goat farms in Northern Jordan's Mafraq Governorate were surveyed using a comprehensive questionnaire to gather data on breeders' information, farm economics, and herd indicators. Dry seasons in arid lands are tough on sheep and goats, with limited fodder and water availability harming their health and reproductive capacity. While wet seasons bring relief, they can also introduce diseases. Special care for these animals throughout the year is crucial to maintain healthy herds in arid regions. Linear models were employed to analyze sample data, with variance and correlation tests used to explore relationships among variables. Additionally, chi-squared analysis was utilized to assess the impact of male replacement plans on these herds. The main outcomes of this study reveal that breeder indicators pose challenges to the sustainability and growth of Awassi sheep and Baladi goat farming. Conversely, economic indicators suggest a blend of financial stability alongside obstacles related to profitability and resource ownership. Breeder age emerges as a significant factor in livestock herd management, influenced by elements such as experience, resource availability, and adherence to traditional practices. Experience notably influences breeders' decisions regarding male insemination replacements, likely due to acquired skills. Crop cultivation is shown to reduce fodder costs, thereby enhancing profitability for each young animal sold and consequently increasing the breeders' returns. Winter conditions elevate animal mortality rates due to cold stress, while other animals within herds contribute to increased kid mortality by facilitating disease transmission. Breeder experience

significantly impacts lamb survival, underscoring its importance in ensuring proper care. Increasing ewe proportions within herds enhances lamb production and lowers lamb mortality rates. Effective ram management and yearling lamb sales further contribute to herd health and productivity. A higher proportion of goats within a herd positively correlates with the number of young yearlings sold but inversely affects breeding males and overall kid numbers, indicating a delicate balance that influences herd productivity and mortality rates. Sheep (92.5%) and goat herds (90.6%) demonstrate reproductive success, indicative of superior long-term vitality. While sheep excel in birth rates, survival, and sales, they exhibit lower ewe and lambing percentages. Goat farming demonstrates high birth rates, sales, and overall success but requires particular attention to ensuring kid survival, especially during the winter months. Improving the characteristics of the Awassi sheep and Baladi goat flocks is crucial for ensuring sustainability, achievable through proper care practices in dry lands. In conclusion, despite the challenges, there are promising opportunities to enhance sheep and goat farming in Jordan. Effective management, informed by the breeders' experience and economics, is crucial. Empowering small-scale herders and implementing targeted interventions, like winter mortality solutions and selective breeding, is vital for sustainable growth.

**Keywords:** sustainability; smallholder; sheep; goat; Jordan

## 1. Introduction

Small ruminant breeding, a global economic mainstay for centuries, faces sustainability threats from climate change and declining genetic diversity [1]. Asia tops the world in raising sheep and goats, with over half a billion each. These animals are important for the continent's economy. While not the global leader, the Middle East plays a significant role in sheep and goat production, which has been deeply ingrained in its social and economic fabric for centuries. Jordan has the following four main goat breeds: Local (Baladi), Damascene (Shami), Black Mountain, and Desert. Their sheep are mostly Awassi, with some Assaf and Najdi breeds [2]. This sector is vital, with approximately 1.4 million sheep and 850,000 goats that are crucial for income, food security, and cultural heritage, especially in remote regions [3–5].

Sheep and goat breeding faces threats from climate change and genetic decline. Empowering smallholder farmers through training and resources is essential, alongside detailed data collection for informed decisions and the preservation of indigenous breeds to ensure genetic diversity. Conservation efforts must be prioritized to secure long-term resilience, making a sustainability assessment imperative [6–8]. In Jordan, sheep and goat farming is crucial for food security, rural development, and cultural heritage, though mainly for meat production. These animals excel in local conditions, are easy to raise, and efficiently utilize roughage. Despite benefits, challenges like diseases and water scarcity constrain sector growth [9–11]. Sustainable approaches offer hope, but environmental and economic hurdles persist [12]. This study underscores the profession's significance for Jordan's economy and rural communities. By identifying challenges, it will inform targeted policies for sustainability enhancement, addressing resource management, market access, and socio-economic development [13,14]. Sustainable sheep and goat farming in arid regions relies on careful herd management, stable feed, genetic selection, proactive healthcare, eco-system preservation, and community involvement. Prioritizing these elements lowers costs, boosts productivity, improves food security, ensures fair benefits, conserves water and biodiversity, combats climate change, and lessens environmental impact [15–17]. To assess farm sustainability, it is crucial to choose suitable tools and indicators tailored to each farm's characteristics and goals [18]. Recognizing diversity among systems is vital for pinpointing areas that need improvement and implementing focused strategies [19]. Enhancing rural goat production is crucial for sector sustainability, utilizing scoring schemes and strategic mating practices to improve production traits and overall farm sustainability [20,21]. Assessing the characteristics of sheep and goat herds in regions

like the Mafraq Governorate is crucial, given their substantial contribution to Jordan's agriculture [22]. Analyzing indicators across socioeconomic and productivity dimensions aims to enhance efficiency and sustainability in sheep and goat farming in specific regions.

This study posits there is an urgent need to focus on developing and strengthening the sheep and goat breeding sector in the dry areas of Jordan. The main challenge is to maintain the sustainability of this sector in light of the increasing climate changes and harsh environmental conditions.

This study aims to evaluate the sustainability of smallholder Awassi sheep and Baladi goat farming in the Jordanian Mafraq Governorate, using multivariate assessment tools. By incorporating various standards related to sustainable production practices, the aim is to enhance the resilience of sheep and goat farming. This comprehensive assessment provides stakeholders with insights into the challenges and opportunities enabling informed decisions to promote sustainable agricultural practices and ensure the prosperity of the rural community in the Mafraq Governorate.

## 2. Materials and Methods

### 2.1. Location and Data

This study was conducted in the northern region of the Jordanian Mafraq Governorate to assess the sustainability of 53 Awassi sheep and Baladi goat farms managed using traditional methods throughout the period of 2021–2022 (Figure 1).

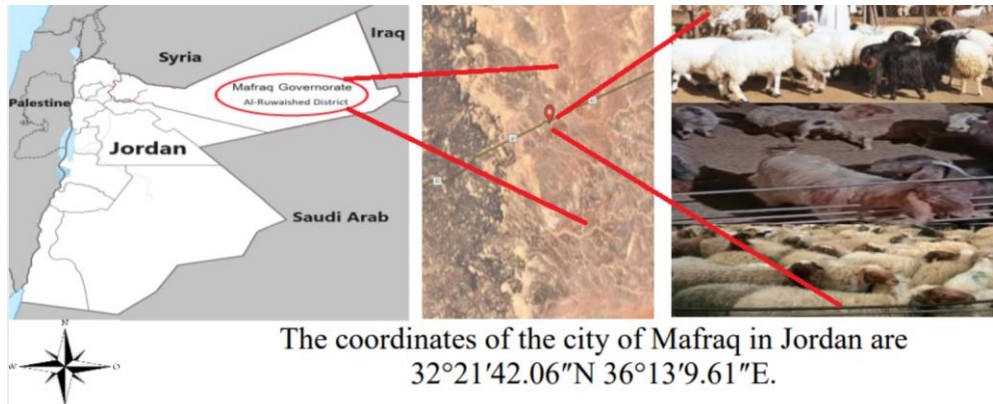

The coordinates of the city of Mafraq in Jordan are 32°21′42.06″N 36°13′9.61″E.

**Figure 1.** On a Jordanian map, indicating the locations of Awassi sheep and Baladi goat herds within the Mafraq Governorate, including the region's coordinates.

The Mafraq Governorate experiences desert conditions with high summer temperatures, low winter temperatures, and moderate rainfall. It grapples with challenges like drought, desertification, and climate change, necessitating effective management of natural resources, and water for sustainable development and environmental protection [23]. Therefore, the study site was selected based on these characteristics.

Sheep and goats play an important role in this sector through meat and dairy production, providing about 49% of the total meat production and 51% of the total dairy production in Jordan (Supplementary Table S1). Awassi sheep and Baladi goats were chosen as study animals due to their significant contribution to the national livestock sector and support for food security and the national economy in Jordan. These animals are characterized by their ability to adapt to harsh climatic conditions, which makes them suitable for breeding in semi-desert areas [24].

The Mafraq Governorate was selected because of the significant sheep and goat population, with approximately 1,548,700 sheep and 22,461 goats in 2021, accounting for 37.4% and 1.90% of the total sheep and goat population in Jordan, respectively (Supplementary Table S2).

Data were collected via a thorough questionnaire designed to gather key details about breeders, farm economics, and herd composition. The information was sourced from farms

registered with the Mafraq Agriculture Directorate, specializing in milk production and livestock. These farms utilize communal pastures and supplement with semi-concentrated feed to meet their herd's fodder requirements.

### 2.2. Statistical Analyses

The methodology employed in this study entailed the utilization of linear models for the analysis of the sample data. To assess the relationships among the variables under investigation, variance and correlation tests were conducted [25]. Specifically, an analysis of variance was carried out to scrutinize total revenues and revenues per animal across both sheep and goat breeders. This analysis adhered to the following linear model (1):

$$Y_{ijklmn} = \mu + LO_i + LC_j + FC_k + B_1 LS_l + B_2 KS_m + e_{ijklmn} \tag{1}$$

where $Y_{ijklmn}$ = the total revenues or revenues/animal of $ijklmn$th observations; $\mu$ = overall mean; $LO_i$ = land ownership (1 = landowner, and 2 = land tenant); $LC_j$ = land cultivation (1 = farmer, and 2 = non farmer); $FC_k$ = fodder cultivation (1 = planting fodder, and 2 = not planting fodder). $LS_l$ = yearling lamb sales%; $KS_m$ = yearling kid sales%; $B_1$ and $B_2$ = coefficients of regression of yearling lamb sales% and yearling kid sales%, respectively; and $e_{ijklmn}$ = random error of $ijklmn$th observations with mean = 0 and variance = $I\sigma_e^2$.

Analysis of variance was used for the flock size of sheep and goat breeders through the following linear model (2):

$$Y_{ijkl} = \mu + EL_i + B_3 EX_j + B_4 AB_K + (EL \times EX)_{ij} + (EL \times AB)_{ik} + (AB \times EX)_{jk} + e_{ijkl} \tag{2}$$

where $Y_{ijkl}$ = the flock size of $ijkl$th observations; $EL_i$ = education level (1 = uneducated, 2 = primary, and 3 = basic); $EX_j$ = experience/years (range 10–30); $AB_k$ = age of breeder/years (range 40–76); $B_3$ and $B_4$ are coefficients of regression of experience and age of breeder, respectively; $(EL \times EX)_{ij}$ = the interaction between education level and experience; $(EL \times AB)_{ik}$ = the interaction between education level and age of breeder; and $(AB \times EX)_{jk}$ = the interaction between the age of the breeder and experience. The symbols in model (1) display the rest.

Analysis of variance was also used for lamb mortality%, kid mortality%, and mortality age/month of sheep and goat herds, following the linear model (3) as follows:

$$Y_{ijklmn} = \mu + OA_i + MM_j + LE_k + B_5 EX_l + B_6 WA_m + e_{ijklmn} \tag{3}$$

where $Y_{ijklmn}$ = the lamb mortality%, kid mortality%, or mortality age of $ijklmn$th observations; $OA_i$ = other animals (1 = other animals exist, and 2 = none, except sheep and goats). $MM_j$ = mortality months (1 = December... May, and 2 = June... November); and $WA_m$ = weaning age/day (range 45–90). The remaining symbols are displayed in the prior models.

Chi-squared analysis examined the male replacement (mating) schedule in a herd of sheep and goats. Spearman's rank correlation coefficient was computed for some economic indicators (Breeder experience, Level of education, Land ownership, Land cultivation, and Fodder cultivation) on the breeders of the sheep and goats. Pearson's correlation coefficient was estimated to explore the relationship between sheep% and goat% herd compositions.

### 2.3. Herd Vitality

To evaluate the vitality of the herds, the study considered the overall reproductive success, a crucial determinant influenced by the birth and survival rates within both sheep and goat herds [26,27]. The reproductive success was quantified as the product of the birth rate and survival rate, encapsulated in Model 4 as follows:

$$Reproductive\ Success = Birth\ rate \times Survival\ rate \tag{4}$$

This equation underscores the fundamental relationship between the birth and survival rates in determining the overall reproductive success of the herds.

*2.4. Sustainability Evaluation*

The indicators studied for herd statistics include sheep%, ewe birth%, lamb%, lamb survival%, and yearling lamb sales%. Similarly, for goats, the metrics comprise goat%, doe birth%, kid%, kid survival%, and yearling kid sales%.

The indicators are categorized into the following three performance levels: "average" signifies standard performance, while "+" denotes performance above average. Conversely, "−" indicates performance below average. A "−" reflects a negative sustainability rating, while a "+" indicates a positive one [28,29].

## 3. Results

*3.1. Breeder Indicators*

The average age of the sheep and goat breeders is 58.72 ± 1.11 years, signaling a lack of interest among younger individuals and posing a sustainability challenge. Family size averages 7.34 ± 0.22 members, with 4.25 ± 0.19 males and 3.09 ± 0.15 females, reflecting a typical structure. Education levels indicate 11.3%, 49.1%, and 39.6% in the uneducated, primary, and basic categories, respectively, revealing a shortage of university education that may affect the future of farming. Breeders have an average of 19.92 ± 0.98 years of experience, emphasizing professionalism.

*3.2. Economic Metrics*

On average, the annual fixed costs amounted to approximately USD 9372.83 ± 268.31, while the variable stood at around USD 791.10 ± 52.16. Total assets and debts were in the range of USD 9108.36 ± 282.41 and USD 265.36 ± 56.03, respectively. Notably, debts represented only 3.33 ± 0.78% of the total assets, underscoring the sustainability of this approach. Furthermore, the annual revenue from sheep and goat farming reached USD 2484.37 ± 93.99, averaging USD 60.91 ± 1.52 per animal. These findings suggest that sheep and goat farming in Northern Jordan holds promise for sustainability. The values were 67.92, 66.04, and 52.83% for sheep and goat farmers who own land, grow crops, and grow green fodder, respectively. These percentages exceed 50% in raising sheep and goats in Northern Jordan; this is a positive indicator. In contrast, the values were 32.08, 33.96, and 47.17% for breeders who rent land, do not grow crops, or do not grow green fodder, respectively.

Table 1 shows that all the correlations for economic indicators were positive ($p < 0.01$). That is, there are strong positive relationships among all indicators, and this indicates that breeders with greater experience tend to have a higher cultural level, land ownership, cultivation of land by fodder, and are better off economically. The strongest association is between land ownership and land cultivation, which indicates that farmers who cultivate land tend to own more land.

**Table 1.** Spearman's correlation coefficients for some economic indicators on the breeders of Awassi sheep and Baladi goats. Prob. > | r | under H0: Rho = 0.

| The Indicators | Level of Education | Land Ownership | Land Cultivation | Fodder Cultivation |
|---|---|---|---|---|
| Experience of breeder/years | 0.89 ** | 0.80 ** | 0.82 ** | 0.86 ** |
| Level of education | | 0.79 ** | 0.83 ** | 0.84 ** |
| Land ownership | | | 0.96 ** | 0.73 ** |
| Land cultivation | | | | 0.76 ** |

**: highly significant. Studied 53 herds (avg. 504 heads/herd) with 283 Awassi sheep and 221 Baladi goats, Jordan.

Crop cultivation significantly enhances ($p < 0.01$; Model 1) the annual returns for sheep and goat breeders, likely attributed to reduced feeding expenses from utilizing crop

residues. Furthermore, the percentage of yearling male lamb sales from ewes ($p < 0.05$; Model 1) underscores the significance of strategic management in maximizing individual animal profitability.

*3.3. Herd Management*

The average total herd size was 503.74 ± 12.66 heads, with Awassi sheep making up 56.48 ± 1.17% and Baladi goats 43.52 ± 1.17%. The weaning age averaged 70.32 ± 1.42 days, with mortality at 8.93 months. 71.70% of the breeders replaced males for insemination every two years, while 28.30% did not. Animal mortality (December–May: 58.49, June–November: 41.49) were studied with rainfall, pasture, and disease in arid regions. The Awassi sheep and Baladi goat herders commonly integrate other animals into their herds, as observed in 73.58% of cases, including camels, horses, chickens, and even dogs and cats, alongside sheep and goats. In Northern Jordan, breeders over 58.72 ± 1.11 years old managed larger herds (7.21 ± 1.12 heads/year), suggesting a positive correlation ($p < 0.05$; Model 2) between breeder age and herd size, possibly influenced by economic, social, or traditional factors. The years of experience as a breeder significantly influenced the decisions regarding male replacements. This likely reflects factors such as accumulated knowledge, skills, and risk tolerance gained over time. Sheep and goat mortality, which is highest in December/January ($p < 0.01$; Model 3), likely due to winter cold, reached 7.7% for lambs and 14.5% for kids. Sharing pens with other animals increases kid mortality ($p < 0.01$; Model 3), possibly due to disease transmission. Breeder experience improves lamb survival ($p < 0.05$; Model 3), highlighting the value of practical care.

Table 2 shows that ewe lambing rates affect lamb numbers, but ram competition can reduce fertility. Good herd management balances growth (selling yearlings) with breeding success (larger herd, lower lamb mortality) for higher lamb sales. Goat herds thrive with more females, but fewer males. Selling yearling kids reduces herd size, so balance is key. Too many males can hurt sales, while disease in kidding goats lowers reproduction and sales. Breeders should support healthcare to boost kid production.

**Table 2.** Pearson's correlation coefficients for Awassi sheep and Baladi goat herd compositions. Prob. > |r| under H0: Rho = 0.

| Sheep | Rams % | NLE % | EL % | AL % | YLS % | LM % |
|---|---|---|---|---|---|---|
| Awassi sheep % | −0.08 | −0.18 | 0.21 | −0.06 | 0.42 ** | −0.50 ** |
| Rams % | | −0.06 | −0.42 ** | −0.44 ** | 0.18 | 0.26 |
| NLE % | | | −0.88 ** | −0.15 | −0.01 | −0.03 |
| EL % | | | | 0.33 * | −0.06 | −0.11 |
| AL % | | | | | −0.75 ** | −0.21 |
| YLS % | | | | | | −0.47 ** |

| Goats | Bucks % | NKD % | KD % | AK % | YKS % | KM % |
|---|---|---|---|---|---|---|
| Baladi goats % | −0.38 ** | −0.03 | 0.17 | −0.33 * | 0.35 ** | 0.07 |
| Bucks % | | 0.15 | −0.56 ** | 0.035 | −0.29 * | 0.043 |
| NKD % | | | −0.90 ** | 0.10 | −0.13 | −0.19 |
| KD % | | | | −0.07 | 0.24 | 0.12 |
| AK % | | | | | −0.06 | −0.77 ** |
| YKS % | | | | | | −0.55 ** |

Non-sign: not significant; *: significant; **: high significant. NLE: Non-lambing ewe%, EL: Ewe lambing%, AL: All lamb%; YLS: Yearling lamb sales%, LM: Lamb mortality%; NKD: Non-kidding doe%, KD: Kidding doe%, AK: All kid%, YKS: Yearling kid sales%, KM: Kids mortality%. Studied herds comprised 283 Awassi sheep and 221 Baladi goats (average herd size 504), Jordan.

According to Table 3, land ownership constitutes the predominant agricultural factor, comprising over 66% of the total. This suggests a link between land ownership and farming, such as sharecropping or tenant farming. The focus on fodder growers indicates an emphasis on livestock rearing. Economic indicators point to an agricultural business with robust economic metrics, including high fixed costs and asset investment, but moderate

profitability with some variability. The herd consists of roughly 56.48% sheep and 43.52% goats. The ewe and doe lambing rates are high at around 95%. This suggests that the flock is likely breeding stock, with a high proportion of females. Moreover, lamb and kid mortality rates are low, at around 3% and 5% respectively, reflecting effective animal husbandry practices. In addition, the weaning age is around 70 days, within the normal range for sheep and goats. Overall, the herds' indicators suggest a well-managed herd with good reproductive performance and low mortality rates.

**Table 3.** Estimated values for Awassi sheep and Baladi goat herds' agricultural, economic, and livestock indicators in drylands of Jordan.

| Agricultural Indicators | | Frequency | Percent (%) |
|---|---|---|---|
| Land ownership | Land owner | 36 | 67.92 |
| | Land tenant | 17 | 32.08 |
| Land cultivation | Farmer | 35 | 66.04 |
| | Non-farmer | 18 | 33.96 |
| Fodder cultivation | grow fodder | 28 | 52.83 |
| | Non-grow fodder | 25 | 47.17 |
| **Metrics and Livestock Indicators** | | **Averages** | **Std Error** |
| Fixed costs/USD | | 9372.83 | 268.31 |
| Variable costs/USD | | 791.10 | 52.16 |
| Activity assets/USD | | 9108.35 | 282.41 |
| year Activity debts/last year/USD | | 265.36 | 56.03 |
| (Activity assets/year Activity debts) $\times$ 100 | | 3.33 | 0.78 |
| Annual return/USD | | 2284.37 | 93.99 |
| Revenue/animal/USD | | 60.91 | 1.51 |
| Awassi sheep% | | 56.48 | 1.17 |
| Male sheep% | | 2.53 | 0.05 |
| Non-lambing% | | 4.59 | 0.10 |
| Ewe birth% | | 95.41 | 0.10 |
| Sheep female% | | 46.44 | 0.06 |
| Lambs% | | 25.34 | 0.29 |
| Lamb sales% | | 18.01 | 0.31 |
| Lamb mortality% | | 3.08 | 0.20 |
| Lamb survival% | | 96.92 | 0.20 |
| Baladi goats% | | 43.52 | 1.17 |
| Male goats% | | 2.76 | 0.06 |
| Non-kidding% | | 4.58 | 0.12 |
| Doe birth% | | 95.42 | 0.12 |
| Goats female% | | 46.33 | 0.07 |
| Kids% | | 23.35 | 0.25 |
| Kids sales% | | 17.91 | 0.21 |
| Kids mortality% | | 5.08 | 0.32 |
| Kids survival% | | 94.93 | 0.32 |
| Weaning age/day | | 70.32 | 1.42 |
| Mortality age/month | | 8.92 | 0.61 |

Studied 53 herds (avg 504) with 283 Awassi sheep and 221 Baladi goats, Jordan.

The breeding program is as follows: The male replacement rate per season is 2.42%. Female replacement rate ranges from 20 to 30%, due to factors like infertility, poor production, aging, or disease. Females begin mating at 7 months and give birth at 12 months. Natural insemination exclusively, with no artificial methods employed.

### 3.4. Sustainability Assessment

Reproductive success was evaluated according to the vitality of the herd. The reproductive success rate in sheep was 92.46 ± 0.181% and 90.57 ± 0.180% in goats (Model 4). This indicates that sheep and goat herds may have a better ability to survive in the long

term than other livestock herds (like cattle) in drylands. However, other factors such as disease resistance, environmental conditions, and management practices also influence herd sustainability. Variations within herds and the specific context must be to obtain a reliable conclusion about herd permanence. Therefore, a more comprehensive analysis that includes additional information is necessary.

Table 4 indicates a mixed sustainability assessment in sheep farming. The sheep% and lamb% receive negative evaluations, but the ewe birth%, lamb survival%, and yearling lamb sales% are above average. This variation in performance across indicators suggests a need for improvements to enhance overall sustainability. The sheep flock is performing well, with some indicators needing additional attention to achieve continuous improvement and increase sustainability.

**Table 4.** Performance indicators for the Awassi sheep within the herds.

| Indicators | Averages | Above Average | Below Average | Sustainability Assessment |
|---|---|---|---|---|
| Awassi sheep% | 56.5 | 47.2 | 52.8 | − |
| Ewes birth% | 95.4 | 60.4 | 39.6 | + |
| Lambs% | 25.3 | 37.7 | 62.3 | − |
| Lamb survival% | 96.9 | 75.5 | 24.5 | + |
| Yearling lamb sales% | 18 | 56.6 | 43.4 | + |

"−" indicates a negative sustainability rating, while "+" indicates a positive one. Studied 53 herds (avg 504) with 283 Awassi sheep, Jordan.

Table 5 shows the sustainability assessment of the goat herd, as sustainability is good in the birth of goats and the percentage of births and their sales. However, it could be bad for the goat herd and the survival of the young. Some indicators must be reviewed for the success of the breeding process. There is a good performance in some indicators, which supports the continued success of goat breeding.

**Table 5.** Performance indicators for the Baladi goats within the herds.

| Indicators | Averages | Above Average | Below Average | Sustainability Assessment |
|---|---|---|---|---|
| Baladi goats% | 43.5 | 45.3 | 54.7 | − |
| Doe birth% | 95.4 | 54.7 | 45.3 | + |
| Kids% | 23.4 | 52.8 | 47.2 | + |
| Kid survival% | 94.9 | 47.2 | 52.8 | − |
| Yearling kid sales% | 17.9 | 69.8 | 30.2 | + |

"−" indicates a negative sustainability rating, while "+" indicates a positive one. Studied 53 herds (avg 504) with 221 Baladi goats, Jordan.

## 4. Discussion

### 4.1. Breeder Indicators

Aging sheep and goat farmers in Jordan lack university education, posing a challenge to sustainability. Attracting younger farmers and improving educational opportunities are crucial to ensure the sustainability of sheep and goat farming in Jordan. This is consistent with the working hypothesis that there is an urgent need to focus on developing and strengthening the sheep and goat breeding sector in the dry areas of Jordan. The sheep and goat farming sector in Jordan and many countries face many challenges, including low profitability due to high production costs and low meat prices, limited grazing land, and a shrinking skilled workforce as young people migrate to urban areas. To address these issues, a collaborative effort between the government, farmers, and civil society is necessary. This collaborative approach could involve implementing incentivizing policies to boost profitability, expanding available grazing areas, improving climate resilience to protect animal well-being, and developing training programs to cultivate a skilled workforce.

Additionally, increased government support, cultural awareness campaigns promoting animal husbandry, and programs specifically designed to encourage youth involvement in the sector are all crucial. By overcoming these challenges, Jordan can strengthen its food security, create new job opportunities, and foster economic growth [30–32].

Some findings of this research are similar to other countries. In Greece, social background, including age, family farming history, experience, and education, significantly affects small ruminant dairy farming [33]. Similar trends hold in Nigeria [34]. India exemplifies regional variations, with sheep and goat farmers typically above 40, often from medium-sized families, and lacking formal education in many cases [35]. These cases highlight the widespread influence of socio-demographic factors within the profession.

*4.2. Economic Metrics*

The Awassi sheep and Baladi goat farming in Northern Jordan appears sustainable. Average annual costs are manageable, debts are low compared to assets, and revenue per animal is positive. Over half of the farmers own land and raise crops or fodder, further strengthening the practice's vitality. Lower annual revenue per animal may be attributed to elevated feeding costs in arid regions, exacerbated by exposure to heat stress in the summer and cold in the winter. Therefore, there is an urgent need to develop and strengthen the sheep and goat breeding sector in the dry areas of Jordan, which is consistent with the hypothesis of this study. Studies on goat production in Mexico, Greece, and the UK highlight the challenges and opportunities in small ruminant farming. While short-term vitality is promising in Mexico [36], long-term sustainability faces hurdles due to high costs. In Greece, factors like labor, loans, feed, education, and age significantly affect goat farmer revenue [37]. Similarly, efficient land use, animal health, and recordkeeping are crucial for UK sheep farm profitability [38]. These studies underline the multifaceted nature of the challenges and opportunities in small ruminant farming globally.

In the literature review, Bundelkhand goats provide a potential income to dryland farmers in India, but face shrinkage and degradation of grazing land. Efforts to restore this land promise a sustainable future [39,40]. Experienced sheep and goat farmers in Jordan are wealthier, often owning land and cultivating crops for their animals. This association between experience and success highlights its importance, with the interesting side note that land ownership usually leads to its cultivation. Ugandan goat farms hold promise for rural lives, dominated by older men with small lands. While goats offer income and cultural value, low earnings and high taxes hold them back. Women manage the production, despite men owning most of the goats. Empowering women/youth is key to unlocking their potential for a brighter future [41]. In New Zealand, sheep production has increased despite declining numbers, highlighting ongoing profitability concerns. Future challenges like changing land use and climate change could affect farmers' profits [42].

*4.3. Herd Management*

In Northern Jordan, the average herd size was 504 heads, predominantly Awassi sheep, with the breeder's age positively linked to herd size. Discontinuing livestock support in Northern Badia halved the animal numbers, leading to price hikes. Proposed solutions included cooperatives, training, and income diversification [43]. In Jordan, sheep and goatherds averaging 297 animals/herd made a USD 24.44 profit with variable costs of USD 18.67 per head [44], this supports the prioritization of sheep and goat breeding in Jordan's arid regions. West African urban areas raise sheep and goats for diverse purposes, valuing sheep more due to location and preference. The sheep breeders there exhibit greater openness to improvement programs, suggesting potential for further gains [45].

The male replacement is likely due to years of experience. Farmers traditionally believe that isolating female sheep or goats triggers ovulation, a phenomenon known as the "male effect", which occurs when replacement males are introduced, rather than isolation itself [46]. This belief aligns with West African practices, which involve frequent male replacement and separate rearing. Incorporating both traditional knowledge and

scientific understanding can enhance animal husbandry practices [47]. In addition, high flock re-placement, driven by demand for lambs rather than ewe age, has been found to reduce production [48].

In Jordan, the winter cold and sharing of pens with other animals increase mortality rates in sheep and goats, especially among newborns. Good animal husbandry practices can lead to healthier young animals. Mortality rates in young small ruminants, notably goats, range from 3.2% to 14.2%, influenced by factors like poor management, age mixing, inadequate vaccination, and low birth weight. This aligns with the working hypothesis advocating for the prioritization of developing and fortifying the sheep and goat breeding sector in Jordan's arid regions.

Most previous research emphasizes improved husbandry practices, proper vaccination, and better care for young animals as key to reducing these losses [49,50]. In addition, the maternal bond of the ewe greatly influences the survival of lambs and is influenced by stress, nutrition, and well-being [51]. Environmental factors also play a role in kid mortality, but proper management can mitigate these challenges [52]. Furthermore, studies reveal variations in disease prevalence across different regions, highlighting the need for targeted control measures [53]. Overall, these studies underscore the multifaceted nature of raising healthy young sheep and goats, highlighting the need for a combination of improved animal husbandry practices, and consideration of environmental factors.

In Jordan, more ewes lambing leads to more yearling lambs sold but fewer kept, suggesting good flock management. A larger flock size reduces lamb mortality, boosting year-round lamb sales. In essence, having more sheep but managing them well can lead to increased yearling lamb sales. No strong correlations were found between specific reproductive traits of ewes and ram sexual performance scores reported by [54]. This suggests that selecting rams based solely on high sexual performance scores is unlikely to improve ewes' reproductive outcomes directly. Low birth weight is linked to a poor environment. Selecting lambs with higher weaning weight promotes growth, survival, and ewe reproduction [55]. Despite maternal influence decreasing with age, selecting larger lambs improves both ewe size and maternal ability due to a positive association, making it a win-win for breeders [56].

For goats, more females mean more kids, while fewer males can increase output. Selling kids reduces herd size. In Jordan, more female goats mean more kids, but selling them shrinks the herd. Balancing the females and sales is crucial, as too many males can harm sales. Keeping goats healthy is essential for kid production.

Heavier kids are born to heavier does, suggesting that feeding goats during critical reproductive periods can improve the health and growth of both dams and offspring [57]. Improving data recording and animal management for the phenotypic selection of traits is an important suggestion [58]. They also emphasize the importance of better environmental conditions for raising kids to enhance survival rates. Positive correlations between the first kidding period and initial milk production in herds suggest that reproductive performance should be considered in the selection criteria [59]. Rainy seasons, older bucks (greater than 5 years old), longer mating periods (at least 21 days), and the presence of multiple males (four or more) were all found to increase birth rates in Mexican goats [60] significantly. High pre-weaning mortality is a significant problem in Gala goats, hindering the development of replacement stock [61]. The study found that the age and sex of the kids were risk factors for mortality. Improved animal health practices could significantly reduce mortality rates and boost overall production in Gala goat herds.

### 4.4. Sustainability Assessment

Herd vitality is important for the sustainability of livestock farming. Healthy and productive herds are more profitable and can better withstand challenges such as drought and disease. The challenge of sustainability in sheep and goat farming is crucial for overall sustainable development. This involves conserving water resources, preserving biodiversity, and ensuring proper animal care. Sustainable practices in these farms contribute to

environmental protection and long-term production continuity while safeguarding natural resources for current and future generations [62,63]. Sustainable sheep and goat farming ensures long-term production by balancing environmental protection with animal well-being and economic vitality. Jordanian sheep and goats appear to thrive, with reproduction rates of 92.5% for sheep and 90.6% for goats, indicating success in dryland conditions. However, sustainability requires thorough checks on disease, environment, and management practices. Therefore, the sheep and goat breeding sector in the dry areas of Jordan must be strengthened, especially in terms of veterinary care and vaccinations.

Intensive production methods can address challenges like heat stress and disease in sheep and goats, aiming for sustainable and ethical farming practices. Achieving this requires collaboration across science, policy, and farming [64]. Compared to cattle, smaller ruminants must adapt to harsh environments, while requiring fewer resources. To enhance their sustainability, prioritizing reducing emissions, ensuring animal welfare, and boosting farm profitability are crucial [65]. Ultimately, sustainable meat production involves utilizing digital tools and embracing agro ecological principles to optimize animal and environmental health [66]. Jordanian sheep farming seems to show mixed results. Furthermore, strong performances in lamb survival and sales still require support for weak spots like overall sheep/lamb numbers. Improvement is crucial for sustainability. Therefore, there is an urgent need to focus on developing and strengthening the sheep-breeding sector in the dry areas of Jordan, and this is consistent with the hypothesis of the study.

Sheep grazing can be sustainable if managed with site-specific densities, minimizing harm to the environment, boosting biodiversity, and protecting ecosystems [67]. A study by [68] identified 37 sustainability indicators that encompass productivity, stability, equity, and all three pillars of sustainability, namely social, economic, and environmental. Economic sustainability related to on- and off-farm herd activities, while social factors like generational turnover and farmer perceptions were key. A trade-off was noted between economic and environmental indicators, indicating that higher economic sustainability often meant lower environmental sustainability. Maintaining genetic diversity is crucial for sustainable sheep and goat farming, according to [69]. Diverse flocks benefit from improved production, reproduction, and adaptability. While traditional breeding methods and modern genomics can be helpful tools, successful strategies would aid in considering the specific context of each farming operation. In Jordan, a three-pronged approach has been proposed to ensure the continued success of the small ruminant sector. This strategy outlined by [70] focuses on developing emergency response plans to mitigate extreme weather events, revamping farmers' skills, and establishing agricultural cooperatives.

In Jordan, goat herd sustainability shows high birth rates and sales but needs review. While breeding success is high, young survival needs monitoring. However, there is an urgent need to focus on developing and strengthening the goat breeding sector in the dry areas of Jordan. Goat farming faces sustainability challenges including climate change, rising fodder and fuel prices, environmental degradation, and genetic erosion [71]. In Africa, sustainable goat farming empowers women, feeds families, and protects the environment, tackling gender inequality, climate change, and rural food insecurity [72]. It is crucial to address marketing and value hurdles faced by Turkey's goat meat to ensure a sustainable future driven by increasing consumer demand [73].

## 5. Conclusions

The Awassi sheep and Baladi goat farming in Northern Jordan show potential for sustainability. Breeders demonstrate good animal husbandry practices with good reproduction rates and low mortality in their herds. Economic indicators show moderate profitability. However, some challenges need to be addressed to achieve long-term sustainability. Educational programs on modern farming techniques can help attract young people to the field. In addition, while lamb and kid mortality rates are low, there is still room for improvement, especially for young goats. An investigation into disease prevention and improvement of veterinary care would be beneficial. Generally, sheep and goat farming in Northern

Jordan shows a positive outlook on sustainability. By addressing identified challenges and leveraging existing strengths, this sector can continue to be a valuable contributor to the Jordanian economy and food security.

This study offers valuable insights into Jordan's sheep and goat profession. Strong animal management practices, land ownership, and moderate profitability can be beneficial in any system, while aging farmers and high newborn mortality rates are challenges that can be addressed through training and veterinary improvements across all production systems. The focus on sustainability emphasizes the potential of all systems to contribute to long-term and stable sheep and goat farming in Jordan.

In addition, this study provides insights into sheep and goat farming. It highlights common challenges, such as attracting young people, and areas for improvement, such as neonatal mortality. Successes in animal husbandry and land management provide standards for other regions. This study also proposes transferable solutions, such as breeding and disease management programs, and provides a framework for evaluating sheep and goat production globally.

**Supplementary Materials:** The following supporting information can be downloaded at: https://www.mdpi.com/article/10.3390/ruminants4020017/s1, Table S1: The population of farm animals and their products; Table S2: Livestock census and production in the Jordanian governorates.

**Author Contributions:** Conceptualization, K.A.-N. and M.D.; Methodology, R.A.T. and F.A.A.-L.; Software, K.A.-N.; Validation, F.A.-B., A.O.K. and A.N.A.-Y.; Formal analysis, A.N.A.-Y.; Investigation, F.A.A.-L.; Resources, F.A.-B.; Data curation, F.A.-B.; Writing—original draft preparation, F.A.-B. and A.O.K.; Writing—review and editing, R.A.T. and F.A.A.-L.; Visualization, A.N.A.-Y.; Supervision, M.D.; Project administration, K.A.-N.; Funding acquisition, M.D. All authors have approved the published manuscript.

**Funding:** This research received no external funding.

**Institutional Review Board Statement:** The study was conducted according to the guidelines of the Declaration of Helsinki and approved by the Institutional Review Board of Jerash University (protocol code z1/1/5 and date of approval 5 December 2022).

**Informed Consent Statement:** Informed consent was taken for this study.

**Data Availability Statement:** The article provides accessible data without farm names mentioned.

**Acknowledgments:** The authors extend their thanks to the Universities of Jerash, Umm Al-Qura, Turku, the National Agriculture Research Center in Amman, and the General Commission for Scientific Agricultural Research in Damascus for their administrative support, which aided in completing this collaborative research.

**Conflicts of Interest:** The authors declare no conflicts of interest.

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
