# Peer review of "Sustainable Sheep and Goat Farming in Arid Regions of Jordan"

_ruminants, doi:10.3390/ruminants4020017_

Round 1
Reviewer 1 Report
Comments and Suggestions for Authors
The article concerns an important issue: sheep and goat breeding in dry areas. The methods and bibliography used seem sufficient. The conclusions are appropriate and have a practical aspect. However, the article needs improvement:
-line 92-93, 105, 108 etc. replace "kid" with "yeanling"
- line 134- "sustainability challenge" - is it really important for the sustainable development of production or simply for the continuity of production? The article does not explain the conditions for sustainable production – line 59-62. The economic, social or environmental element of this particular production was not explained. Please explain the full scope of what sustainable production means at the beginning of the article.
Please pay attention to the use of the word "sustainable" throughout the work - whether it is necessary and whether it results from the content.
This also applies to the title, because sustainable development of breeding includes an economic aspect - there is no point in emphasizing it separately.
- line 148 - please add the national to international currency conversion rate, e.g. US dollar or euro
- line 154- "sustainable source of income" - one could talk about sustainable income by comparing it over time - there is no analysis over time - please add such an analysis or remove it. An element could be added comparing the income from this production with other types of animal or, for example, plant production carried out in this region.
- table 3, 4, 5 - what is the abbreviation S.O.V.? all abbreviations that are used in the text should be explained the first time and then the abbreviations can be used.
Best regards
Author Response
Please, see the attachment.

Reviewer 2 Report
Comments and Suggestions for Authors
Referee’s Evaluation Report
MANUSCRIPT IDENTIFICATION:  RUMIN-2952177
 
Challenges, economics, and sustainability of sheep and goat farming in drylands
 (ORIGINAL ARTICLE - A SURVEY ARTICLE)
 
Comments to Authors/Editor:
The paper of Al-Barakeh & colleagues aimed to quantify the sustainability of smallholder sheep and goat farming in Jordan throughout a multivariate approach tool based on criteria about sustainable production. This manuscript falls within the scope of RUMINANTS. The manuscript is sufficiently informative for the replication of the study.  In general, the organization of the study seems to be well designed, yet the English quality, grammar, and sentence structure is certainly fragile; it must be greatly improved. The Abstract section was written in a careless fashion. The abstract must start with the research question, the aim and importance of the study, the location (i.e., LW, LN), the methodology, the sampling size, as well as the main analysis approach. Then, the main results, including numbers and probabilities, to finish with the main conclusions. The abstract must be rewritten. Regarding the Introduction section, both grammar and structure are certainly fragile (i.e., L44-49). Are the authors comfortable with this opening paragraph in the introduction section??? The authors must contextualize their study adding information regarding the inventory, along with the social and economic importance of sheep and goat production in Asia, the Middle-East and Jordan. While the objectives of the study were clearly stated, no working hypothesis of the study was proposed; this is a must. Please include the contribution of sheep and goats to the national livestock sector; why do the authors use small-ruminant production instead of, for instance, bovine???  Why the authors selected the Mafraq Governorate location instead another region??? Regarding the M&M section, please start this section with “Location of the study and environmental conditions”. In general, the M&M section includes information relevant and in accordance with the general idea of the study. As commented, the English quality must be improved also in this section; L77, “live meat”!!!!!!; what does it means??? L85, variables using??? Why in the whole Jordan, was this region selected to perform the study??? In general, the methods used are relevant and in accordance with the general idea of the study. L83; Statistical analysis or Statistical Analyses???, correct accordingly. Please divide the Result and Discussion section to: 3. Result and 4. Discussion. In the actual Result and Discussion section format, it is difficult to understand the whole idea of the study. Certainly, with this R&D format, there is a lack of fluidity in the information, and that is a handicap for the readership of RUMINANTS. In the Results section, please include in Tables all the information obtained from the study, and in the writing part of Results, only talk about the main outcomes obtained from the analyses. Regarding the econometric analyses, please include the currency exchange from JD to euros or US dollars; remember that the article is directed to an international readership. L153-154, compared to what??? L170-171 & L182, &L184; do not include both the words “statistically significant” and the probability level itself; it is a pleonasm, use only the p-value. L171-172; ..”between all indicators , and this indicates”…, use a different word; it is cacophony. In general, the authors never used any p-value in the text; this is a must. L199, what kind of animals??? What is the idea to include the ANOVA, instead of the average per variables??? Remove Table 3; since the age of breeder was the only response variable depicting significance, just mention in the paragraph that the others did not differ between classes and only include the general average for each one; what is the idea to include the mean square??? The only response variables that differed among classes were Experience (yr), “other animals???” (%) and mortality (mo.). Please, also eliminate Tables 4 and 5, and only include the general averages for each variable non depicting differences among classes. L278-279; any statistical difference between sheep and goats??? The observed difference was 92.5 vs. 90.s, less that 2%; do that suggests to the authors long-term viability???, are you serious??? Where is the statistical evidence to support such a statement??? Please reform Tables 7 & 8; invert the axis. The novelty value of the results is from fair to fragile. In all Tables, the titles must be rewritten; remember that any Table must stand by itself. The way in which the authors presented the Result and Discussion section was very unfortunate; it loss fluidity and scientific merit. Certainly, there is a lack of connection and fluidity among such section and the results reported in the scientific global literature; the authors must homogenate the presentation of the Results and then, in a different section, align them in the Discussion section. Besides, I really do suggest to the authors initiate the Discussion of their main research outcomes including the working hypothesis of the study. Authors must define if, with the obtained results, such a hypothesis is rejected or non-rejected. For this reason, the authors must include the working hypothesis prior to the objectives in the Introduction section. Besides, the authors must link, in a logical fashion, their main findings in the Discussion section; they need to compare & discuss and, at some point, be able to propose the main explanations for such specific outcomes, considering previous similar studies from the scientific literature. In fact, the authors confronted, in a limited fashion, their main research outcomes with other studies from the global scientific literature; please solve this limited approach. A crucial point to solve that, is to present in one section the Results and, as I mentioned, the Discussion section as one. In general, the authors made an accurate interpretation of the main findings. Besides, most of the Tables are non-friendly from a visual standpoint. Moreover, most “variable names” are not defined neither in the title nor in a footnote; this is nonacceptable. Please correct along with all the Tables this unfortunate situation. With respect to the Conclusion section, the authors must highlight the main findings of the study and the possible use of the study outcomes upon sheep & goat production in the different regions or production systems in Jordan and try to extrapolate to a global perspective. Please remember that Conclusions are not a Mini-Abstract. From my point of view, Conclusion MUST be rewritten; it must be aligned with the working hypothesis and main results; therefore, the working hypothesis MUST be included in the Introduction section. The list of references cited in the manuscript is proper, while actualized. This is a fair study, with an interesting set of response variables. Yet, on one side the authors must improve both the English language quality as well as the clarity and logical arrangement of the whole manuscript; the authors must use an orthodox format to present the manuscript; Results in one section and Discussion in another one. It is central to ensure that the paper is readable; the authors must increase the readability and the scientific writing and merit of the manuscript. All the commented issues and requests must be clearly addressed by the authors; at this point, and based on the above comments, my pronouncement is that this manuscript cannot be accepted in its actual format.  It requires extensive editions and corrections.
Comments on the Quality of English Language
As mentioned, the authors must increase the readability and the scientific writing and merit of the manuscript.
Author Response
Please, see the attachment.

Reviewer 3 Report
Comments and Suggestions for Authors
Authors in their manuscript entitled "Challenges, Economics, and Sustainability of Sheep and Goat Farming in Dryland" present valuable information in regard to zootechnical and economic indices of sheep and goat farming in Jordan. The study refers to a specific location therefore I recommend authors to include this in the abstract as well as in the title since there is not a full coverage for results output of Dryland areas as a whole. In addition, there are some points that need further clarification or improvements before any recommendation for further publication. These points are as follow and I sense that will non dufficult the authors.
Figure 1. Legend must be more precise: i.e.Location,coordinates and typical herds of sheep and goats incuded in the study
l 85 using-->used
l 103. The meaning is not clear please rephrase
l 107 specify E*Xi
l 13 absence of youths. Please discuss further this issue. why this happening in your country and compare with other countries
l. 137 positive sign for sustainability with the increase in age. It seems somehow controversial taking together the absence of youths to continue in the sector as well as the low education level. Please explain better
l. 142. Only in Greece? What is happening in your country or other countries. Please compare.
l. 153 42,64 JD per year? Isn't it quite low?
l 199 other animals? you mean other species? Please specify which species.
l.206 what do you mean with "requiring effective strategies"? The previous mentioned solutions why aren't they effective?
Table 4 specify p values
l 214-215 meaning is not so clear. please rephrase
l. 224 this is not supported by the data presented. Please present the relative results
248. "No strong associations were found between specific ewe reproductive traits and ram sexual performance scores". This derives from the present study or previous studies? It is not clear.
Results: I recommend to authors to present a table (even as supplementary) with all values estimated for each farming unit regarding zootechnical and economical indexes as well as a mean for each parameter. In addition it is not clear if authors present reproductive parameters like i.e. replacement rates for males and females, first age of mating (male/female), first age of birth and if artificial insemination is used.
Discussion. The authors should elaborate a little more in making comparisons between their results and results coming from similar production systems or other countries emphasizing in the specific fetures of dry land environments.
Comments on the Quality of English Language
Some parts of the manuscript need further rephrasing by the authors to make clear the meaning they want to point out.
Author Response
Please, see the attachment.

Round 2
Reviewer 2 Report
Comments and Suggestions for Authors
Referee’s Evaluation Report
MANUSCRIPT IDENTIFICATION:  RUMIN-2952177-R1
 
Sustaining sheep and goat farming in arid regions of Jordan
 (ORIGINAL ARTICLE - A SURVEY ARTICLE)
 
Comments to Authors/Editor:
This is the R1 version of the paper of Al-Barakeh & colleagues that aims to quantify the sustainability of smallholder sheep and goat farming in Jordan throughout a multivariate approach tool based on criteria about sustainable production. In general, the organization of the study, the English quality, grammar, and sentence structure in the R1 version has been improved. The title was modified; the new title is more accurate with respect to the aims of the study; excellent decision. The Abstract section was certainly improved; please, according to the scientific basic rules, do not personalize the study; L22-23, --In 2021.2022, the researchers surveyed…???!!!, are the authors comfortable with this writing style???; try: In 2021-2022, a total of 53 traditional sheep and goat farms in…., were surveyed. L25; herd viability???, what does this mean??; define.., “In this study, heard viability was defined as (move L28-29, to L25…”); this is central for the study to be defined as early as possible in the abstract. L27; between variables or among variables??? L30; This research finds???; are the authors comfortable with this writing??; try, The main outcomes of the study indicates that breeder…. In this R1 version, the authors neither mentioned in the Abstract the main sheep and goat breeds, nor indicated if these breeds show reproductive seasonality, and, if so, how this seasonality can affect the “Herd viability”; please disclose some ideas on this issue. L36; selling lambs yearling; what does this mean???; please rephrase. L44-45, … but low sheep and lamb %; what does this mean??; sheep or ewe??, lamb% or lambing % ??? L45; Improvement is needed for sustainability, and the potential is high???; what does this mean??; please rephrase. L46; … and other sustainability issues…, such as what???? This section was improved, but it certainly must be rewritten. Sorry about that; it lacks clarity and several non-sense sentences must be clarified. Regarding the Introduction section, both grammar and structure are still fragile (i.e., L59; social and economic fabric???). In this R1 version, the authors certainly contextualized their study by adding information regarding the inventory, along with the social and economic importance of sheep and goat production in Asia, the Middle East, and Jordan. Yet, in L62; .. a sizable inventory…, include the inventory!!!!! The idea presented in L54-56, is repeated in LL68-69; I suggest eliminating L54 to 56, start the Introduction with L57. L72; equip them???; it is not the best “word choice”; rephrase. L74-75 is repetitive with L83-85; please adjust accordingly. L87; environmental well-being???; please rephrase. L90; if goats are preferred, why Jordan has more sheep than goats????; it makes no sense at all!!!!; correct accordingly. The use of almost 115 lines for the Introduction section is just an excess; the authors must use no more than 3 paragraphs in the Introduction section, to state the scientific question or the gap of information regarding the topic of the study, contextualize the scientific question from a social, economic and production stand, project the working hypothesis and the objectives of the study; that is it, no more than that. Correct accordingly. Besides, the authors never stated why they selected the Mafraq Governorate location instead another region; please move to the Introduction section some ideas presented from L175-181. There are a lot of repeated ideas from L96 up to L168, correct-adjust accordingly; the Introduction must be shortened. In general, the M&M section of this R1 version includes information relevant and in accordance with the general idea of the study. As commented, the English quality was certainly improved but it needs to be revised again. Fortunately, this R1 version includes the Results as a sole section, now is easier to understand what the authors try to explain. L291, Economic metrics; the authors must define the currency expressing the presented amounts (i.e., United States Dollars, USD). L299; rewrite. L314 & 315, please adjust or restructure the “yearling” concept-idea; in the actual format is extremely confusing. As requested in the previous review ELIMINATE those Tables which include the ANOVA; it is not necessary to include such information, it is just “visual contamination”. The authors already explained in depth the statistical models in the M&M section. Certainly, an old-fashion strategy was to include such information both, in the M&M and Results section, but nowadays is quite uncommon. By the way, in the answer from the authors to reviewers, the authors developed a short dissertation regarding the importance of ANOVAS, Square means, probabilities; believe me, a know about statistics (i.e., more than 170 JCR-published articles), and, nowadays, it makes no sense at all to include such an information, especially when you use very basic statistical analyses such as ANOVAS, correlations (Sperman & Pearson) as well as Chi Square approaches. To include such information does not generate neither a more robust manuscript nor add any more scientific merit. It is better to use time, effort, and knowledge to explain the Results and discuss them. Please remember, the readership of the manuscript has limited time to read any article; they only occasionally will read any manuscript to the last full stop, MOST will not. Therefore, our job as scientists is to make sure that whatever they do read, we will expose them to the MOST important information that we have to OFFER, and that they will RETAIN it. Do not be upset with my comments or suggestions; I only try to present to you, the authors, some ideas that would help to present a better product to the readership. How you will retain the attention of the readership???, well, as I always comment to my graduate students, there are four key words that may help to prepare and present a good scientific manuscript: structure, logic, readability, and economy. The reader´s mind is not a sponge, they can only take in and retain some key information, and believe me, to include the ANOVA, the square roots and other superfluous information is not important at all!!! So, please eliminate that information from the main body of the manuscript!!!!! You can include such information, if that make you happy, in supplementary archives, along with the data itself. L320, the authors include for the first time the Awassi breed; at least the three main breeds of sheep, and the main three breed of goats, must be included as early on as in the Abstract section!!!! L326-327, …, animal mortality rate in December until May was 58.49 compared to 41.49 in June until January; why the authors divide the mortality analysis is such periods??, what is the biological base of that, if any?? Moreover, how is possible that both classes include moths from each other???, the first includes dec, jan, feb, mar, apr & may, while the other-one includes jun, july, aug, sep, oct, nov, dec and jan!!!???, That is, both periods include December and January, therefore, I can expect confounded effects from such comparisons. What is the rationale behind the conformation of such periods??? As previously requested, eliminate Table 3, 4, & 5, and move to the “Supplementary information” and include the original data in Excell format. Table 6 is ok. In Table 7 and 8, how may breeds of sheep and goats were included in the analyses??? For this reason, the authors must include the number of breeds per specie, and the number of animals per breed within herd-specie, otherwise, I can expect, again, confounded effect in such comparisons. If the authors are trying to establish the sustainability of sheep and goats under day-land conditions, the authors are obligated to include the type of breeds included in the analyses. Then, the authors must define, from the breeds-specie analyzed, what breed depicted the best performance under harsh climatic conditions. Table 9 is ok. In general, the methods used are relevant and in accordance with the general idea of the study. The novelty value of the results is from fair to fragile. Again, as requested, in all Tables, the titles must be rewritten; remember that any Table must stand by itself. As mentioned, the way in which the authors presented the Result section with all that information of ANOVAS and Square Roots was certainly unfortunate; it loss fluidity and scientific merit. Certainly, there is a lack of connection and fluidity when presenting the obtained results. As I requested previously, the authors must initiate the Discussion of their main research outcomes including the working hypothesis of the study. Authors must define if, with the obtained results, such a hypothesis is rejected or non-rejected. Fortunately, the authors included the working hypothesis in this R1 version. As suggested, the authors must link, in a logical fashion, their main findings in the Discussion section; they need to compare & discuss and, at some point, be able to propose the main explanations for such specific outcomes, considering previous similar studies from the scientific literature. With respect to the Conclusion section, the authors must highlight the main findings of the study and the possible use of the study outcomes upon sheep & goat production in the different regions or production systems in Jordan and try to extrapolate to a global perspective. Please remember that Conclusions are not a Mini-Abstract. Although improved in this R1 version, from my point of view, the Conclusion section MUST be shortened while aligned with the working hypothesis and main results. The list of references cited in the manuscript is proper, while actualized. This is a fair study, with an interesting set of response variables. While this R1 version was certainly very improved, the authors still must do some major adjustments regarding the clarity and logical arrangement of the whole manuscript. Again, all the commented issues and requests must be clearly addressed by the authors; at this point, and based on the above comments, my pronouncement is that this manuscript cannot be accepted in its actual format; this R1 version still requires extensive editions and corrections.
Comments on the Quality of English Language
No comments other than those already mentioned...
Author Response
Please, see the attachment.

Reviewer 3 Report
Comments and Suggestions for Authors
Authors have made extensive improvemnet of their manuscript quility and also addressed all the issues noted by my comments. I can, therefore, reccomend the further acceptance of the manuscript.
Author Response
About your comment "The authors have made extensive improvements to the quality of their manuscript and have also addressed all of the issues pointed out by my comments. I can therefore recommend further acceptance of the manuscript.
Sincerely, we thank you for taking the time to review this manuscript. Your efforts are greatly appreciated. Please refer to the revised research version.